# Hyperthermic Intraperitoneal Chemotherapy (HIPEC): New Approaches and Controversies on the Treatment of Advanced Epithelial Ovarian Cancer—Systematic Review and Meta-Analysis

**DOI:** 10.3390/jcm12227012

**Published:** 2023-11-09

**Authors:** Luigi Della Corte, Carmine Conte, Mario Palumbo, Serena Guerra, Dario Colacurci, Gaetano Riemma, Pasquale De Franciscis, Pierluigi Giampaolino, Anna Fagotti, Giuseppe Bifulco, Giovanni Scambia

**Affiliations:** 1Department of Neuroscience, Reproductive Sciences and Dentistry, School of Medicine, University of Naples “Federico II”, 80131 Naples, Italy; giuseppe.bifulco@unina.it; 2Department of Woman, Child and Public Health, Fondazione Policlinico Universitario A. Gemelli IRCCS, 00168 Roma, Italy; carmine.conte@policlinicogemelli.it (C.C.); anna.fagotti@policlinicogemelli.it (A.F.); giovanni.scambia@policlinicogemelli.it (G.S.); 3Department of Public Health, School of Medicine, University of Naples “Federico II”, 80131 Naples, Italy; mario.palumbo@unina.it (M.P.); serena.guerra@unina.it (S.G.); dario.colacurci@unina.it (D.C.); pierluigi.giampaolino@unina.it (P.G.); 4Department of Woman, Child and General and Specialized Surgery, University of Campania “Luigi Vanvitelli”, 80138 Naples, Italy; pasquale.defranciscis@unicampania.it

**Keywords:** HIPEC, epitelian ovarian cancer, intraoperative chemotherapy, interval debulking surgery, recurrent ovarian cancer, primary debulking surgery

## Abstract

Background: Hyperthermic intraperitoneal chemotherapy (HIPEC) after cytoreductive surgery has been extensively studied in patients with peritoneal carcinomatosis, and it holds promise as a therapeutic strategy, but its role remains elusive. The aim of this study was to assess the existing evidence for the use or not of HIPEC in primary debulking surgery (PDS), interval debulking surgery (IDS), and recurrent ovarian cancer (ROC), evaluated in terms of survival rates and post-surgical morbidity. Methods: Medline, Pubmed, Cochrane, and Medscape were systematically searched for any article comparing the use of HIPEC treatment with any other therapy in patients with ovarian cancer in PDS, IDS, and ROC. Preferred Reporting Items for Systematic Reviews and Meta-Analyses (PRISMA) reporting guidelines were followed. We only considered English-language published studies. Results: We included 14 studies, including two RCTs with a total of 1813 women, published between 2003 and 2023 with a recruitment period between 1998 and 2020. In PDS, there were no differences in progression-free survival (PFS) between HIPEC and controls [MD −5.53 months [95% CI −19.91 to 8.84 months]; I^2^ = 96%]. Conversely, in patients treated with NACT, pooled results showed a significant survival advantage in terms of progression-free survival (PFS) and overall survival (OS) in the combined HIPEC plus IDS group rather than surgery alone [PFS: MD 4.68 months (95% CI 3.49 to 5.86 months, I^2^ = 95%); OS: MD 11.81 months (95% CI 9.34 to 14.27 months); I^2^ = 97%]. Concerning ROC patients, pooled MD did not show either a significant PFS difference between intervention and controls [MD 2.68 months (95% CI 433 to 9.70 months); I^2^ = 95%], and OS significant difference (MD 6.69 months [95% CI −9.09 to 22.47 months]; I^2^ = 98%). Severe post-operative complications (≥grade 3) were available in 10 studies, accounting for 1108 women. Overall, there was a slightly but significantly increased risk with the combined approach compared to controls [RR 1.26 (95% CI 1.02 to 1.55); I^2^ = 0%]. Conclusions: The combination of HIPEC with cytoreductive surgery prolongs OS and PFS in advanced epithelial ovarian cancer after NACT with acceptable morbidity. However, additional trials are still needed to determine the effectiveness of HIPEC in primary and recurrence settings. In the era of personalized medicine, the correlation between the efficacy of HIPEC and biological and molecular findings represents a challenge for the future of ovarian cancer.

## 1. Introduction

Ovarian cancer is the third most common gynaecological malignancy in the world after cervix uteri and corpus uteri neoplasms but the most lethal (incidence-to-mortality ratio, IMR: 0.63 ovary vs. 0.55 cervix uteri vs. 0.21 corpus uteri) [1]. Ovarian cancer is most frequently diagnosed among women 55–64 years old (mean age at diagnosis: 63) [2]. Approximately 70% of women with epithelial ovarian cancer (EOC) are diagnosed with advanced disease (FIGO stage > II, i.e., tumour extending out of the pelvis), which is associated with high post-operative morbidity and a worse prognosis [3]. In cases of regional or distant disease at diagnosis, the 5-year survival rate is below 40% [2].

The most effective first-line treatment for advanced ovarian cancer is primary debulking surgery (PDS) aimed at maximal cytoreduction (no visible disease) or optimal cytoreduction (one or more cancer residues measuring up to 10 mm), followed by adjuvant intravenous chemotherapy based on platinum and taxanes, according to tumour sensitivity. An alternative treatment option is interval debulking surgery (IDS) with the goal of maximum cytoreduction, performed after neoadjuvant chemotherapy (NACT), typically consisting of 3 or 4 cycles of upfront chemotherapy.

Recurrence in epithelial ovarian cancer occurs in approximately 75% of women within two years from the first diagnosis, according to stage and primary treatment [4]. In the case of recurrent ovarian cancer (ROC), the standard treatment includes systemic chemotherapy, whereas the role of surgery is still under discussion. Secondary cytoreductive surgery (SCS) is an option for selected patients recurring after PDS or NACT/IDS [5]. Selection criteria include platinum sensitivity, potentially resectable disease, previous complete resection of cancer, localized disease, and absence of ascites in patients with good performance status. Two randomized phase III trials (AGO DESKTOP III and SOC-1) have compared the benefit of SCS plus second-line chemotherapy versus chemotherapy alone in ROC [5,6,7]. Results from the randomized DESKTOP-III trial have demonstrated an overall survival advantage in patients undergoing SCS followed by chemotherapy, compared with chemotherapy alone, mainly when complete cytoreduction is achieved [5].

Patients with single or oligometastatic recurrences can be offered minimally invasive secondary cytoreductive surgery, mainly if localized in the lymph nodes, and/or receive neoadjuvant chemotherapy at primary diagnosis [8,9,10,11,12,13]. Hyperthermic intraperitoneal chemotherapy (HIPEC) after cytoreductive surgery has been extensively studied in patients with peritoneal carcinomatosis, and it holds promise as a therapeutic strategy in both primary and recurrent scenarios. The intraperitoneal route directly delivers chemotherapeutic agents into the peritoneal cavity, allowing higher doses and greater in-site concentration with lower systemic side effects than the intravenous route. Also, hyperthermia enhances the pharmacokinetics and pharmacodynamics of chemotherapy agents used, increasing drug absorption and boosting cytotoxic effects [14,15,16,17].

Only a few Phase III prospective comparative studies have tested whether HIPEC improves outcomes for patients with advanced ovarian cancer [12]. Despite the established rationale and these encouraging results, a certain degree of scepticism still surrounds HIPEC in advanced ovarian cancer, involving inherent potential morbidity and the paucity of randomized data confirming its theoretical advantage [18]. To date, HIPEC is not recommended for patients undergoing PDS while waiting for the results of an ongoing international randomized phase III trial enrolling patients with newly diagnosed FIGO stage III epithelial ovarian cancer. This trial (M06OVH-OVHIPEC) has shown HIPEC leading to improvement in recurrence-free survival and overall survival in patients with FIGO stage III primary epithelial ovarian, fallopian tube, or peritoneal cancer who underwent NACT and IDS, and it did not result in higher rates of side effects [12]. 

Thus, HIPEC has been incorporated into the National Comprehensive Cancer Network (NCCN) guidelines for patients with FIGO stage III ovarian cancer at the time of IDS in patients with stable disease or a response to neoadjuvant chemotherapy [19,20,21,22]. Among the patients with ROC, only one randomized controlled trial has evaluated the effect and safety of HIPEC [20,21]. The authors reported that HIPEC resulted in survival benefits for patients with ROC, but there are different limitations considering the randomization process and the definition of the endpoints [21]. In a recent meta-analysis, HIPEC was associated with better OS (hazard ratio = 0.566) but not with PFS in ROC patients. However, HIPEC improved PFS in patients with residual tumors ≤1 cm or no visible tumors, while it improved OS for only those with ≤1 cm [14]. This study aims to add to the existing evidence an overall outline of the role and benefits of HIPEC in EOC with peritoneal carcinomatosis in PDS, IDS, and ROC, evaluated in terms of survival rates and post-surgical morbidity.

## 2. Materials and Methods

The methods for this study were specified a priori based on the recommendations in the Preferred Reporting Items for Systematic Reviews and Meta-Analyses (PRISMA) statement. We obtained registration to the International Prospective Register of Systematic Reviews (PROSPERO) database for this paper (ID no. CRD42023442437).

### 2.1. Search Method

The Medline, PubMed, Cochrane, and Medscape databases were systematically searched for any article comparing the use of HIPEC treatment with any other therapy in patients with ovarian cancer in PDS, IDS, and ROC. This study was reported in accordance with the Preferred Reporting Items for Systematic Reviews and Meta-Analyses (PRISMA) reporting guidelines. Authors only considered English-language published studies.

### 2.2. Study Selection

Two independent investigators (L.D.C. and S.G.) screened all studies identified in our search by titles and abstracts for eligibility. Any article identified as having the potential to fulfill our inclusion criteria underwent full-text evaluation. If agreement on eligibility was not reached between the two investigators, a third investigator (M.P.) was involved to evaluate the article. The eligibility was defined by the PICO Framework: Population (P): patients diagnosed with ovarian cancer undergoing PDS, IDS, or recurrent disease treatment; Intervention (I) Hyperthermic Intraperitoneal Chemotherapy; Comparison: people without HIPEC treatment; Outcomes (O): overall free survival, progression-free survival, complications.

### 2.3. Data Extraction

The data extraction was carried out by two authors (L.D.C. and C.C.), who filled out a pre-piloted extraction form independently. Any disagreement was resolved by consensus. Multiple records reporting on the same trial were excluded. In the case of double reporting data in conference abstracts and article publications, only the data from the publication were considered. The data extraction included: first author, country and year of publication, recruitment period, study population, EOC Stage (FIGO), type of tumour (Serous, others), BRCA mutations, HIPEC protocol details, median overall survival (OS), median progression-free survival (PFS), rate of restricted Performance Status (ECOG ≥ 1), rate of bowel resections, and rate of grade ≥ 3 adverse events occurrence after surgery (according to the National Cancer Institute Common Terminology Criteria for Adverse Events, version 4.0).

### 2.4. Outcomes

The primary outcome of interest is the OS of patients with primary, advanced, and ROC treated with HIPEC versus treatment without HIPEC, evaluated as the mean difference (MD) expressed in months between intervention and controls. Secondary outcomes are disease-free survival (DFS) for primary and PFS for ROC measured as mean difference (MD) expressed in months between intervention and controls, as well as post-operative performance status between 0 and 2 according to ECOG criteria, postoperative complications, and morbidity.

### 2.5. Statistical Analysis

Review Manager 5.3 (The Nordic Cochrane Centre 2014) and STATA, version 14.1 (StataCorp LLC, College Station, TX, USA), were used to analyse the data. After using Der Simonian and Laird’s random-effects model, the summary measures were presented as a risk ratio (RR) or MD with a 95% confidence interval (CI). We utilized a Higgins I^2^ index greater than 0% to address any possible heterogeneity. The effects of type of surgery combined with HIPEC subtypes (primary debulking, interval debulking, or recurrent surgery) on main and secondary outcomes were examined using subgroup analysis. Using a visual inspection of the funnel plot, the possible publication bias was investigated. Statistical significance was defined as a *p*-value less than 0.05.

The Engauge Digitizer v. 4.1 program was used to extract survival information from the Kaplan–Meier curves for studies in which the related findings were not displayed.

### 2.6. Quality Assessment

The quality of the included studies was assessed using the Newcastle–Ottawa scale (NOS) for observational cohort studies [17] and with the criteria expressed in the Cochrane risk of bias tool in the case of randomized controlled studies. This assessment scale uses three broad factors (selection, comparability, and exposure), with scores ranging from 0 (lowest quality) to 9 (best quality). Two authors (L.D.C. and M.P.) independently rated this study’s quality. Any disagreement was subsequently resolved by discussion or consultation with a third author (G.R.). The NOS criteria and their related scores are reported in Appendix A.

## 3. Results

### 3.1. General Characteristics

This study analyzed data for advanced and ROC patients with stages IIIA–IV, according to FIGO. Figure 1 reports the flow chart of the inclusion and exclusion of papers available in the literature according to PRISMA guidelines.

The reports included were published between 2003 and 2023, with a recruitment period between 1998 and 2020. In this analysis, we included studies where information about all patients was easily found, such as type of surgery, type of chemotherapy, HIPEC technique, and Overall Survival and Progression Free Disease. The patients analyzed come from studies that performed cytoreductive surgery followed or preceded by chemotherapy. We preferred studies that confronted HIPEC vs. HIPEC plus surgery and excluded those that did not respect the inclusion criteria previously described. We analyzed three types of patients: patients who undergo primary cytoreductive surgery followed by HIPEC; patients who undergo IDS after cycles of NACT followed by HIPEC; and patients with ROC who undergo HIPEC. Most of the studies included a control branch that did not perform HIPEC.

The primary outcomes analyzed are the Progression Free Disease (PFS), i.e., the number of months between the therapy and the recurrence of the disease, and the Overall Survival, i.e., the number of months between the therapy and the death of the patient. In most of the studies analyzed, cisplatin was the main drug used. Carboplatin [23,24], Paclitaxel [25,26], and Doxorubicin [20] were also used. The studies differ for the HIPEC protocol in duration, temperature, and dosage of the drug. The histology did not meet inclusion criteria; however, high-grade serous ovarian cancer is the most common. Refs. [24,27] gave information about the BRCA mutation of the patient.

### 3.2. Quality Assessment

The NOS and the Cochrane Tool were used to evaluate the studies’ quality, and the results indicated a generally positive score for the case-control studies’ ascertainment of the important outcomes as well as for the selection and comparability of this study groups. A low risk of bias was found in 6 out of 7 of the assessed items for the RCT [14] (Appendix A).

### 3.3. Survival Outcomes

A quantitative overview of OS and PFS according to the type of ovarian surgery is depicted in Table 1.

#### 3.3.1. Primary Debulking Surgery

Progression Free Survival (PFS) after PDS was evaluated in three studies [22,27,34]. The general characteristics of primary debulking surgery patients are described in Table 2.

HIPEC protocol characteristics after primary debulking surgery are described in Table 3.

Ghirardi et al. [22] reported data for two different cohorts (BRCA-wild type and BRCA-mutated women) for 177 women. Overall, there were no differences in PFS between HIPEC and controls (MD −5.53 months [95% CI −19.91 to 8.84 months]; I^2^ = 96%) (Figure 2). OS was only reported in one study [27] with 584 patients, showing an increased mean OS (MD 16.70 months [95% CI 16.30 to 17.10 months]) in patients undergoing HIPEC compared to no HIPEC.

#### 3.3.2. Interval Debulking Surgery

General characteristics of interval debulking surgery patients and HIPEC protocols are described in Table 4 and Table 5.

PFS after combined HIPEC and IDS was retrieved in eight studies (911 women) [12,18,20,27,29,30,31]; Pooled results showed a significant advantage in the combined HIPEC plus IDS group rather than surgery alone (MD 4.68 months in comparison with the other data [21,22,23,24,25,26,28,29,33,35]) reported (95% CI 3.49 to 5.86 months; I^2^ = 95%) (Figure 3).

Concerning OS, data regarding eight studies and a sample of 909 people were retrieved, showing a significant difference between groups favoring the combined approach [MD 11.81 months (95% CI 9.34 to 14.27 months); I^2^ = 97%]. (Figure 4) [12,16,18,27,29,30,31].

#### 3.3.3. Recurrent Ovarian Cancer

Five studies accounting for 285 women evaluated the PFS differences among HIPEC and no-HIPEC administration in ROC [14,19,33]. The general characteristics of ROC patients are described in Table 6. The HIPEC protocol characteristics of ROC patients are described in Table 7 and Table 8.

Pooled MD did not show a PFS significant difference between intervention and controls [MD 2.68 months (95% CI −4.33 to 9.70 months); I^2^ = 95%] (Figure 5).

Meanwhile, OS was reported in 2 trials (218 women) [14,19]. Of those, Spiliotis et al. [14] reported data separately according to platinum resistance and sensitivity [14]. Overall, pooled results did not report a significant difference between the two approaches in terms of OS (MD 6.69 months [95% CI −9.09 to 22.47 months]; I^2^ = 98%) (Figure 6).

### 3.4. Surgical Outcomes

Table 9 depicts the main outcomes for this quantitative analysis.

Complications referred at least to grade 3 according to the Clavien–Dindo classification were available in 10 studies, accounting for 1108 women undergoing HIPEC for primary, IDS, or secondary cytoreductive surgery for ovarian cancer relative to non-HIPEC controls [12,19,21,22,24,28,29,32,33,35]. There was a slightly but significantly increased risk with the combined approach compared to controls [RR 1.26 (95% CI 1.02 to 1.55); I^2^ = 0%] (Figure 7).

Similarly, a slightly but significant reduced risk for a performance status between 0 and 2 was seen when HIPEC was not involved in the surgical protocol [RR 0.63 (95% CI 0.40 to 0.99); I^2^ = 0%], according to 6 studies [20,21,22,24,28,32] and 1083 women (Figure 8).

Detailed surgical complications for HIPEC and non-HIPEC groups are reported in Table 10, Table 11 and Table 12.

An increased risk for renal complications was seen in women undergoing HIPEC plus surgery for ovarian cancer relative to surgery alone [RR 1.28 (95% CI 1.03 to 1.59); I^2^ = 0%], while no differences were discovered for the remaining outcomes (Table 10).

## 4. Discussion

The interest in HIPEC is growing in current literature because of the typical peritoneal spread of ovarian cancer. The addition of HIPEC to cytoreductive surgery for the treatment of ovarian cancer is feasible, but its efficacy is under investigation in patients with (almost) complete gross resection. The use of HIPEC is based mainly on retrospective data and a few heterogeneous RCTs; consequently, the value of HIPEC in cytoreductive surgery for ovarian cancer remains controversial in most guidelines. This meta-analysis focused on patients who underwent either PDS and IDS for advanced ovarian cancer or SCS for recurrence, and we analyzed these groups separately. In the analysis, we could identify an association between the value of HIPEC at CRS for EOC and the best timing for this treatment. These results confirmed the results obtained in the previous RCT [12], reporting the efficacy of HIPEC administration in the IDS group in terms of both PFS (MD −5.53 months) and OS (MD 11.81 months). Van Driel [12] reported the first RCT with evidence of HIPEC’s survival benefit in advanced EOC after NACT; therefore, its application has been introduced as an option in the NCCN guidelines [13]. The hazard ratio (HR) for disease recurrence or death was 0.66 (95% CI 0.50–0.87, *p* = 0.003), favoring the HIPEC group. The median PFS was 14.2 months in the CRS plus HIPEC group versus 10.7 months in the CRS group. At 5 years, 50% of the patients in the CRS plus HIPEC group had died versus 62% in the CRS group (HR 0.67, 95% CI 0.48–0.94, *p* = 0.020). The median OS was 45.7 months versus 33.9 months.

However, this study has been criticized despite being an RCT. In particular, the long-term nature of this study (9 years), the timing of randomization as a possible surgical bias (patients were randomized before the surgical procedure), the shorter PFS than anticipated (more than 6 months in both arms, even though approximately 70% had a surgically complete resection), and the high rate of ostomy in the HIPEC group (72% vs. 43% in the control arm) Similar positive survival outcomes in this setting were documented in the other 2 works, with the limitation of small sample groups [20,21,27]. The reason why the use of HIPEC after NACT provides some benefit may arise from the possibility of patients/drug selection in the neoadjuvant phase. Only chemosensitive patients will receive HIPEC, with drug sensitivity having already been tested in the neoadjuvant regimens.

For primary surgery and the use of HIPEC, there is the most heterogeneous data.

Ghirardi et al. [22] reported no differences in PFS between HIPEC and controls [MD −5.53 months [95% CI −19.91 to 8.84 months]; I^2^ = 96%], but Lim et al. [27] showed an increased mean OS (MD 16.70 months [95% CI 16.30 to 17.10 months]) in patients undergoing HIPEC. Additional trials are still needed to determine the optimal timing for HIPEC administration [30,36] and whether HIPEC is also effective after primary cytoreductive surgery in prospective randomized trials [37,38,39]. Mendivil et al. [18] showed a significant PFS advantage in the HIPEC group for PDS but no advantages in OS. However, the control group was recruited much earlier (2008–2014) and thus had a longer median follow-up time in contrast to the HIPEC group, which was collected from 2012–2015. Also, the Italian group of Ceresoli et al. [29] reported a lesser peritoneal recurrence after HIPEC treatment. This difference in relapse pattern seems to affect the OS (no median reached vs. 35.5 months (*p* = 0.048)), with better results in patients treated with HIPEC.

In this setting of patients, Ghirardi et al. demonstrated the survival benefit of BRCA wildtype (wt) patients compared to BRCA mutated patients [22]. The authors hypothesize that HIPEC may balance the decreased chemosensitivity of BRCA-wild-type patients compared to BRCA-mutated patients. Indeed, in the control arm (no HIPEC), a significantly higher difference in both PFS (*p* = 0.011) and OS (*p* = 0.003) in BRCA-mutated patients compared to BRCA-wild-type patients was demonstrated. Among the patients with ROC, there is evidence that even in relapsed EOC, CRS can improve prognosis, provided that the tumor can be resected completely [5,7]. Only two randomized controlled trials have evaluated the effect and safety of HIPEC, and they reported contradictory results [20,24].

In particular, Spiliotis et al. [14] showed a higher median OS in the HIPEC group compared to the SCS group (26.7 months vs. 13.4 months, respectively; *p* = 0.006) [20]. The authors did not report data on PFS and toxicity profiles, further limiting the interpretation of the data. The RCT by Spiliotis et al. [14] was the only one that also included platinum-resistant patients, and it demonstrated that the overall survival rates were the same in the HIPEC cohorts regardless of the presence or absence of platinum resistance. Still, this result is influenced by good recurrence removal to prolong median overall survival. However, this study has been heavily criticized for its validity and the scientific value of its findings, so any conclusions from this study have limited clinical relevance. The second RCT trial [24] failed to demonstrate an improved survival in patients receiving HIPEC (with carboplatin 800 mg/m^2^ for 90 min) compared to patients undergoing SCS only, with median OS 52.5 vs. 59.7 months, respectively (*p* = 0.310), and median PFS 12.3 and 15.7 months, respectively (*p* = 0.05 with significance stated at *p* < 0.05). In this study, we have some limitations, such as the inclusion criteria of only patients with a maximum PFI of 30 months and no PARP inhibitor administration as maintenance therapy.

Also, in the platinum resistance debate for PDS, Lei et al. [28] performed a cohort study from 2010 to 2017 to compare survival outcomes between PDS with HIPEC and PDS alone for patients with stage III EOC. A total of 584 patients with stage III primary EOC were treated with either PDS alone or PDS with HIPEC. The median follow-up period was 42.2 (33.3–51.0) months. Incomplete tumor mass resection without HIPEC exhibits the worst outcome, with a median survival of 19.9 months (95% CI, 11.6–39.1) and a 3-year overall survival rate of 36.7% (95% CI, 23.4–50.1%). In patients with R0 resection with additional HIPEC, median survival was 53.9 months (95% CI, 46.6–63.7), and 3-year overall survival was 65.9% (95% CI, 60.1–71.2%). Patients with a residual tumor who underwent HIPEC therapy had a median survival of 29.2 months (95% CI, 22.3–45.5) and a 3-year overall survival rate of 44.3% (95% CI, 34.6–53.4%). For patients with complete tumor mass reduction who received PDS only, median survival was 42.3 months (95% CI, 31.1–59.3), and 3-year overall survival was 55.4% (95% CI, 44.7–64.8%).

Further RCTs on relapsed EOC need to be performed, as the literature is very heterogeneous regarding results, first-line postoperative treatment, and median follow-up. The salient point of the literature is the safety profile of HIPEC. We need to understand whether the use of the drug exceeds the threshold between tolerability and toxicity. Related to the toxicity of HIPEC administration, in this meta-analysis we reported a slightly but significant increased risk of severe (≥grade 3) postoperative complications with the combined approach (cytoreduction plus HIPEC) compared to controls (no HIPEC) [RR 1.26 (95% CI 1.02 to 1.55); I^2^ = 0%] in the overall analysis.

In the PDS setting, Ghirardi et al. [22] showed no significant differences were detected between the two populations regarding early postoperative complications (*p* = 0.920). Among patients who experienced severe (G3–G4) early postoperative complications, 4 and 1 patients belonged to the HIPEC and no HIPEC groups, respectively. Regarding the patients who underwent NACT before cytoreduction, Van Driel et al. [12] reported adverse events of grade 3 or 4 in 30 patients (25%) in the surgery group and in 32 patients (27%) in the surgery-plus-HIPEC group (*p* = 0.76). In both groups, the most common events in grades 3 or 4 were abdominal pain, infection, and ileus. The reported data on recurrence patients is controversial. Zivanovic et al. [13] did not show any difference in perioperative mortality, use of ostomies, length of stay, or postoperative toxicity between the two groups. Baiocchi et al. [33] reported a higher rate of Grade 3–4 complications in the HIPEC group (34.5%) vs. the control group (10.6%) (*p* = 0.015). However, there were no perioperative deaths (within 30 days after surgery) in the HIPEC group and 2 (4.0%) deaths in the SCS group.

The major criticism in terms of toxicity between the studies is the different chemotherapeutic agents used, the different anesthesiologic protocols, the different settings of populations (probably different extents of surgery), and the long HIPEC experience. The analysis of the complications shows a statistically significant difference between the group treated with HIPEC compared to the control group (RR = 1.26, 95% CI = 1.02–1.55, *p* = 0.03). However, the content of this assessment is influenced by two studies [24,33] which are not statistically significant in terms of complications, and one with a smaller sample heterogeneity than others [29]. The results are reflected in data from the literature, suggesting that the rate of post-operative morbidity and mortality may be correlated with the experience of the clinical centers that perform the technique [40,41]. Thus, in experienced centers, HIPEC should be considered safe both for clinical practice and research applications. Regarding OC recurrence, a randomized phase III trial (CHIPOR), opened in April 2011 and currently ongoing, reported a significant improvement in OS and peritoneal PFS with the first platinum-sensitive relapse of EOC treated with second-line platinum-based chemotherapy followed by secondary complete cytoreductive surgery [42].

There are some limitations existing in the analysis of the data published so far: the inclusion criteria and HIPEC drug regimens for EOC are different concerning the extent of disease status and CRS, and therefore, it is difficult to state a standard quantitative measurement of the morbidity related to HIPEC. Overall, the insufficient RCT data promotes the accumulation of bias. The attempt at an overall interpretation of the most important studies in the literature is the strength of our work, but at the same time, the limits of our analysis are related to the data themselves and their interpretation.

A common concern in the HIPEC debate for EOC treatment is that none of the currently used drugs have been developed for intraperitoneal administration, and we do not know all the effects of these on the integrity of the mesothelium; indeed, their administration is one and the duration of therapy is limited, and compared to these limits, we demand ourselves what the real effect of the therapy is [41]. Data from in vitro experiments reported that the response of cancer to drugs depends on cell cycle phase, cell cycle time, drug concentration, and duration of treatment, which should be improved by the use of prolonged delivery formulations with hydrogels, according to recent studies [41,43]. Furthermore, all drugs administered during HIPEC are off-label; it is very challenging to find the thin threshold between obtaining better outcomes on one side and reducing the rate of complications or unexpected effects on the other [41,43].

Regarding the most appropriate chemotherapeutic agent, cisplatin 100 mg/m^2^ is recommended in the NCCN Guidelines, but paclitaxel represents another possibility. Other studies focus on the choice of de-escalation of cisplatin to 75 mg/m^2^ and the use of bevacizumab and poly (ADP-ribose) polymerase (PARP) inhibitors for maintenance and immunotherapy [44].

Another point of discussion concerns the method of administration. In this regard, prospective studies are lacking, but the few data present in the literature confirm that there are no significant differences between “open technique” with the wall layers not closed and “closed abdomen perfusion” with the skin and/or fascial layer closed to prevent gaseous diffusion into the environment and reduce the risk of infection [45].

The relevant inconsistencies among the studies published so far are related to different factors such as performance status, comorbidities, and/or age of enrolled patients or the chance of complete cytoreduction, as well as the skills of the surgeon and the possibility of having a multidisciplinary team in case of intra-/post-operative complications [46]. HIPEC data for IDS seems robust; those on PDS have to be confirmed by OVHIPEC-2 [11].

Further trials are needed, especially for recurrent EOC; precision medicine may certainly provide support in this regard in the future [47].

## 5. Conclusions

The combination of HIPEC with standard surgery prolongs OS and PFS in advanced EOC after NACT without more complications, and a complete gross resection increases this prolonged survival.

The same results in terms of safety and efficacy are mostly lacking in PDS and relapsed patients, and new data are awaited from ongoing randomized trials.

Moreover, in the era of precision medicine, future analyses are needed to tailor the HIPEC administration according to BRCA mutational status in the different ovarian cancer settings, integrating into the trials the analysis of PARP inhibitor maintenance treatment (possible synergistic effect?).

In the ovarian cancer scenario, the HIPEC administration remains uncertain, but these recent results are encouraging given the use of a loco-regional treatment to treat a typically loco-regional diffusion of the disease. We await further evidence to define the role of HIPEC therapy in treating EOC, and the use of analyses such as these is essential to pushing the scientific world in this direction.

## Figures and Tables

**Figure 1 jcm-12-07012-f001:**
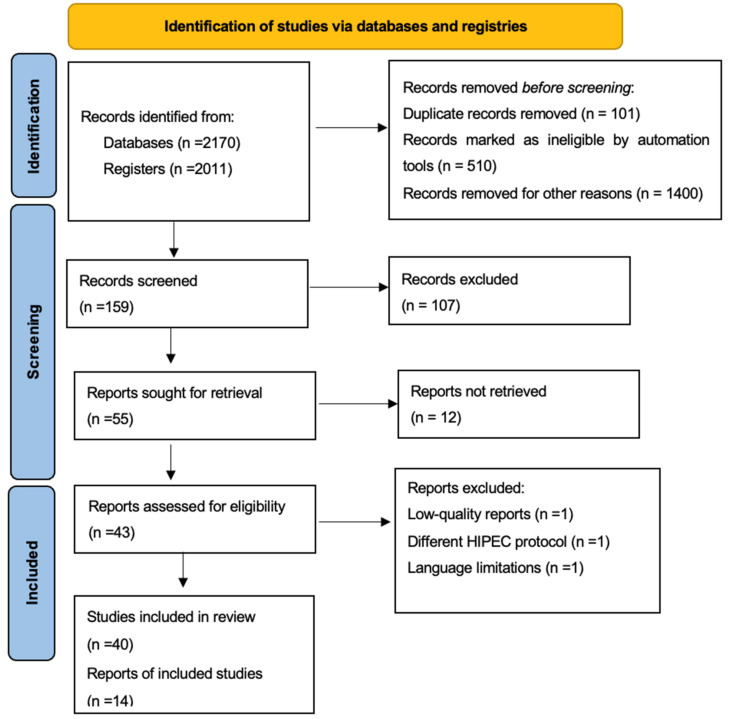
Flow chart of included and excluded studies.

**Figure 2 jcm-12-07012-f002:**
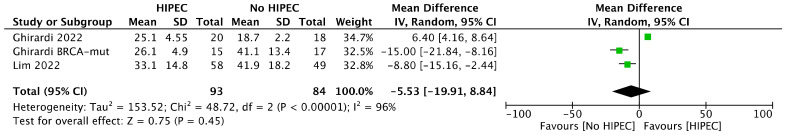
Forest plot for Progression Free Survival (PFS) in Primary Debulking Surgery.

**Figure 3 jcm-12-07012-f003:**
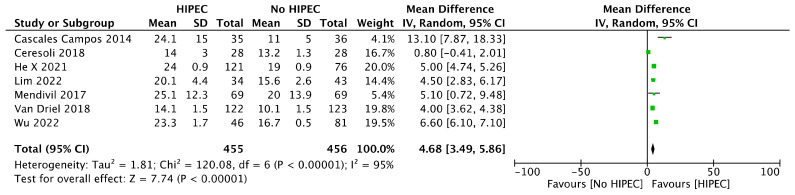
Forest plot for Progression Free Survival (PFS) in Interval Debulking Surgery.

**Figure 4 jcm-12-07012-f004:**
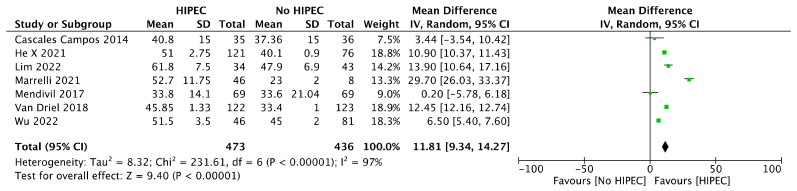
Forest plot for Overall Survival (OS) in Interval Debulking Surgery.

**Figure 5 jcm-12-07012-f005:**
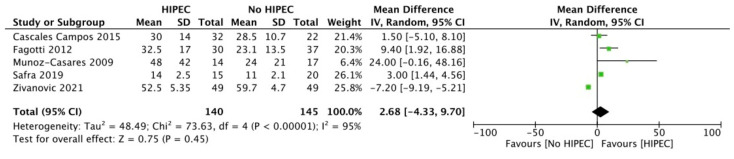
Forest plot for Progression Free Survival (PFS) in Recurrent Ovarian Cancer.

**Figure 6 jcm-12-07012-f006:**
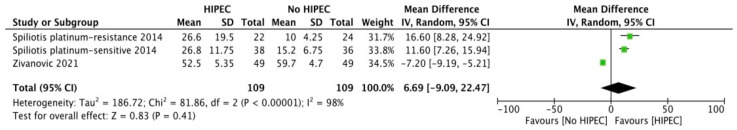
Forest plot for Overall Survival (OS) in Recurrent Ovarian Cancer.

**Figure 7 jcm-12-07012-f007:**
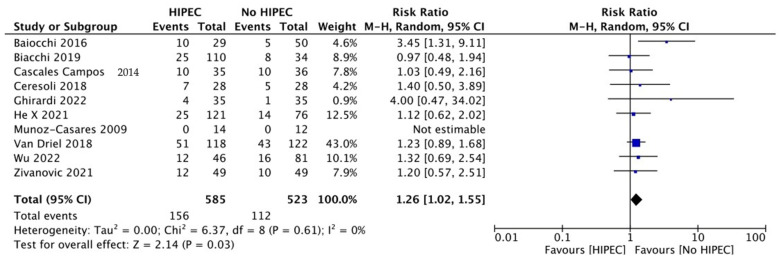
Forest plot for Grade 3 Complications (Clavien–Dindo).

**Figure 8 jcm-12-07012-f008:**
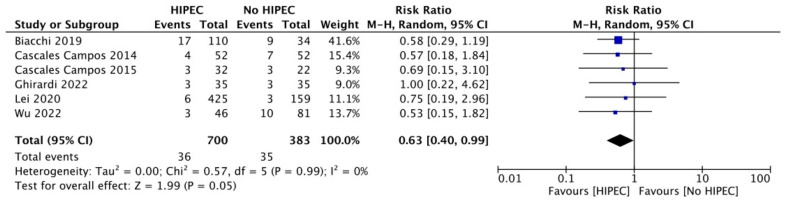
Forest plot for Performance Status (PS) according to ECOG.

**Table 1 jcm-12-07012-t001:** Quantitative characteristics of survival outcomes.

Author	Year	Median OS (Months)		Median PFS (Months)	
		HIPEC	Controls	HIPEC	Controls
Primary Debulking Surgery					
Lei [28]	2020	49.8	34.0	NR	NR
Lim [27] (PDS group)	2021	71.3	NR	23.9	29.7
Ghirardi [22]	2022	NR	NR	NR	NR
BRCA-wild type		NR	62.0	26.6	19.4
BRCA-mutated		NR	NR	26.7	38.0
Interval Debulking Surgery					
Mendivil [18]	2017	33.8	33.6	25.1	20.0
Ceresoli [29]	2018	NR	32.5	14	13.2
Van Driel [12]	2018	45.7	33.9	14.2	10.7
Cascales Campos [20]	2014	52	45	18	12
Batista [30]	2021	NR	NR	18.1	NR
Marrelli [31]	2021	53	23	22	NR
Lim [27] (IDS group)	2022	61.8	48.2	17.4	15.4
Wu [32]	2022	51	44	22	16
Recurrent Ovarian Cancer					
Spiliotis [14]	2014	NR	NR	NR	NR
Platinum sensitive disease		26.8	15.2	NR	NR
Platinum resistance disease		26.6	10.2	NR	NR
Cascales Campos [21]	2015	NR	NR	21	22
Baiocchi [33]	2016	58.3	59.3	15.8	18.6
Zivanovic [19]	2021	52.5	59.7	12.3	15.7

NR: not reported.

**Table 2 jcm-12-07012-t002:** Primary debulking surgery general characteristics.

Author	Country	Year	Recruitment Period	EOC Stage (FIGO)	HIPEC Group, No. (%)	Control Group, No. (%)
Ghirardi [22]	Italy	2022	2010–2015	IIIB-IV	35 (50%)	35 (50%)
Lei [28]	China	2020	2010–2017	III	425 (73.46%)	159 (26.54%)
Lim [27] (subgroup PDS)	Republic of Korea	2022	2010–2016	III–IV	58 (54.2%)	49 (45.8%)
Cascales Campos [20] (subgroup PDS)	Spain	2014	1998–2011	IIIC-IV	23 (44%)	20 (57%)

**Table 3 jcm-12-07012-t003:** Primary debulking surgery follows the HIPEC protocol.

Author	Year	HIPEC Protocol			Type of Tumor: Serous, No. (%)		Type of Tumor: Others, No. (%)		BRCA Mutations, No. (%)		No BRCA Mutations (HIPEC-No HIPEC)	No Tested BRCA Mutations (HIPEC-No HIPEC)
		HIPEC Drug	Temp (°C)	Duration (min)	HIPEC	Controls	HIPEC	Controls	HIPEC	Controls		
Ghirardi [22]	2022	Cisplatin 75 mg/m^2^	41.5	60	NR	NR	NR	NR	15/35 (42.9%)	17/35 (48.6%)	20/35 (57.1%)–18/35 (51.4%)	NR
Lei [28]	2020	Cisplatin 50 mg/m^2^	43	60	419/425 (98.6%)	156/159 (98.1%)	6/425 (1.4%)	3/159 (1.8%)	NR	NR	NR	NR
Lim [27] (PDS subgroup)	2022	Cisplatin 75mg/m^2^	41.5	90	53/58 (91.4%)	41/49 (83.7%)	5/58 (8.6%)	8/49 (16.3%)	NR	NR	NR	NR
Cascales Campos [20] (PDS subgroup)	2014	Cisplatin 75 mg/m^2^	42–43	>60	NR	NR	NR	NR	NR	NR	NR	NR

NR: not reported.

**Table 4 jcm-12-07012-t004:** Interval debulking surgery general characteristics.

Author	Country	Year	Recruitment Period	EOC Stage (FIGO)	HIPEC Group, No. (%)	Control Group, No. (%)
Batista [30]	Portugal	2021	2015−2019	IIIB–IV	15	0
Cascales Campos [20] (IDS subgroup)	Spain	2014	1998−2011	IIIC–IV	29 (56%)	15 (45%)
Ceresoli [29]	Italy	2018	2010−2016	IIIC–IV	28 (50%)	28 (50%)
Lim [27] (IDS subgroup)	Republic of Korea	2022	2010−2016	III−IV	34 (44%)	43 (66%)
Marrelli [31]	Italy	2021	2007−2014	III	46 (85%)	8 (15%)
Mendivil [18]	Germany	2017	2012−2015	IIIA–IIIB–IIIC–IV	69 (50%)	69 (50%)
Wu [32]	China	2022	2012−2020	IIIC–IV	46 (36.2%)	81 (63.8%)
Van Driel [12]	The Netherlands	2018	2007−2016	III	122 (49.8%)	123 (50.2%)

HIPEC protocol characteristics after IDS are described in Table 4.

**Table 5 jcm-12-07012-t005:** Interval debulking surgery follows the HIPEC protocol.

Author	Year	HIPEC Protocol			Type of Tumor: Serous, No. (%)		Type of Tumor: Others No. (%)		BRCA Mutations, No. (%)		No BRCA Mutations (HIPEC-No HIPEC)	No Tested BRCA Mutations (HIPEC-No HIPEC)
		HIPEC Drug	Temp (°C)	Duration (min)	HIPEC	Controls	HIPEC	Controls	HIPEC	Controls		
Batista [30]	2021	Cisplatin (25 mg/L)—cisplatin + doxorubicin (15 mg/L)	41–43	30	12/15 (80%)	0	3/15 (20%)	0	4/15 (26%)	0	NR	NR
Cascales Campos [20] (IDS subgroup)	2014	Cisplatin 75 mg/m^2^	42–43	>60	NR	NR	NR	NR	NR	NR	NR	NR
Ceresoli [29]	2018	Cisplatin 100 mg/m^2^—paclitaxel 175 mg/m^2^	41.5	90	25/28 (89.3%)	27/28 (96.4%)	3/28 (10.7%)	1/28 (3.6%)	NR	NR	NR	NR
Lim (IDS subgroup) [27]	2022	Cisplatin 75 mg/m^2^	41.5	90	32/34 (94.1%)	38/43 (88.4%)	2/34 (5.9%)	7/43 (11,6%)	NR	NR	NR	NR
Marrelli [31]	2021	Mitomycin C 25 mg/m^2^—cisplatin 100 mg/m	41–42	60	37/46 (80%)	NR	9/46 (20%)	NR	NR	NR	NR	NR
Mendivil [18]	2017	Carboplatin AUC10	41.5	90	48/69 (69.6%)	45/69 (65.2%)	21/69 (30.4%)	24/69 (34.8%)	NR	NR	NR	NR
Wu [32]	2022	Cisplatin 70–80 mg/m^2^	43	60	46/46 (100%)	81/81 (100%)	0/46 (0%)	0/81 (0%)	8/46 (17.4%)	17/81 (21%)	19/46 (41.3%)–45/81 (55.6%)	19/46 (41.3%)–19/81 (23.5%)
Van Driel [12]	2018	Cisplatin 100 mg/m^2^	40	90	116/122 (95%)	109/123 (88.6%)	6/122 (5%)	14/123 (11.4%)	NR	NR	NR	NR

NR: not reported.

**Table 6 jcm-12-07012-t006:** Recurrent ovarian cancer has general characteristics.

Author	Country	Year	Recruitment Period	EOC Stage (FIGO)	HIPEC Group, No. (%)	Control Group, No. (%)
Spiliotis [14]						
Platinum sensitive disease	Greece	2014	2006–2013	IIIC–IV	38 (51%)	36 (49%)
Platinum resistant disease	Greece	2014	2006–2013	IIIC–IV	22 (47.8%)	24 (52.2%)
Cascales Campos [21]	Spain	2015	2001–2012	I–IV	32 (59%)	22 (41%)
Baiocchi [33]	Brazil	2016	2000–2014	I–IV	29 (36.7%)	50 (63.3%)
Zivanovic [19]	Germany	2021	2014–2019	I–IV	49 (50%)	49 (50%)

**Table 7 jcm-12-07012-t007:** Recurrent ovarian cancer HIPEC protocol.

Author	Year	HIPEC Protocol			Type of Tumor: Serous, No. (%)		Type of Tumor: Others No. (%)	
		HIPEC Drug	Temp (°C)	Duration (min)	HIPEC	Controls	HIPEC	Controls
Spiliotis [14]								
Platinum sensitive disease	2014	Cisplatin 100 mg/m^2^ + paclitaxel 175 mg/m^2^	42.5	60	NR	NR	NR	NR
Platinum resistant disease	2014	Doxorubicin 35 mg/m^2^ + paclitaxel 175 mg/m^2^	42.5	60	NR	NR	NR	NR
Cascales Campos [21]	2015	Paclitaxel 60 mg/m^2^	42	NR	NR	NR	NR	NR
Baiocchi [33]	2016	Mitomycin C 10 mg/m^2^—Cisplatin 50 mg/m^2^—Doxorubicin	41–42	90	18/29 (62%)	38/50 (76%)	11/29 (38%)	12/50 (24%)
Zivanovic [19]	2021	Carboplatin 800 mg/m^2^	41–43	90	47/49 (96%)	*NR*	48/49 (98%)	NR

NR: not reported.

**Table 8 jcm-12-07012-t008:** Recurrent ovarian cancer BRCA mutation.

Author	Year	No. (%) No BRCA Mutations (HIPEC-No HIPEC)	No. (%) BRCA Mutations (HIPEC-No HIPEC)
Cascales Campos [20]	2014	NR	NR
Spiliotis [14]	2015	NR	NR
Platinum sensitive disease	2014	NR	NR
Platinum resistant disease	2014	NR	NR
Cascales Campos [21]	2015	NR	NR
Baiocchi [33]	2016	NR	NR
Zivanovic [19]	2021	39/49 (80%)–38/49 (78%)	10/49 (20%)–11/49 (22%)

NR: not reported.

**Table 9 jcm-12-07012-t009:** Outcomes for HIPEC vs Control groups in available studies.

Author	Year	≥Grade 3 Complications, No. (%)		Restricted Performance Status (ECOG ≥ 1), No. (%)	
		HIPEC	Controls	HIPEC	Controls
Primary Debulking Surgery					
Lei [28]	2020	NR	NR	NR	NR
Lim (PDS group) [27]	2021	NR	NR	NR	NR
Ghirardi [17]	2022	4/35 (11.42%)	1/35 (2.8%)	3/35 (8.6%)	3/35 (8.6%)
BRCA-wild type		NR	NR	NR	NR
BRCA-mutated		NR	NR	NR	NR
Interval Debulking Surgery					
Mendivil [18]	2017	0/69 (0%)	NR	NR	NR
Ceresoli [29]	2018	7/28 (25%)	5/28 (18%)	NR	NR
Van Driel [12]	2018	51/118 (43%)	43/122 (35%)	NR	NR
Cascales Campos [20]	2014	NR	NR	NR	NR
Batista [30]	2021	3/15 (20%)	0	12/15 (80%)	0
Marrelli [31]	2021	13/46 (28%)	NR	NR	NR
Lim (group ICS) [27]	2022	NR	NR	NR	NR
Wu [32]	2022	12/46 (26%)	16/81 (20%)	3/46 (6.5%)	10/81 (12%)
Recurrent Ovarian Cancer					
Spiliotis [14]	2014	NR	NR	NR	NR
Platinum sensitive disease		NR	NR	NR	NR
Platinum resistant disease		NR	NR	NR	NR
Cascales Campos [21]	2015	NR	NR	NR	NR
Baiocchi [33]	2016	10/29 (34.5%)	5/50 (10.6%)	NR	NR
Zivanovic [19]	2021	12/49 (24%)	10/49 (20%)	NR	NR

NR: not reported.

**Table 10 jcm-12-07012-t010:** Risk for complications between HIPEC and non-HIPEC procedure.

	Studies	Participants	RR (95% CI)	I^2^
Anemia	5 [12,13,27,28,34]	1100	1.02 (0.98 to 1.05)	5%
Bowel related complications	5 [12,21,24,25,28]	665	1.14 (0.66 to 1.99)	70%
Dyspnea	5 [12,21,27,28,34]	1111	1.01 (0.67 to 1.51)	0%
Renal complications	3 [27,28,34]	802	1.28 (1.03 to 1.59)	0%
Haemorrhage	5 [19,21,27,28,34]	1052	0.62 (0.18 to 2.08)	31%

**Table 11 jcm-12-07012-t011:** HIPEC vs. no HIPEC complications (part 1).

Author	Year	Bowel related Complications (HIPEC-No HIPEC)	Anemia (HIPEC-No HIPEC)	Dyspnea (HIPEC-No HIPEC)	Hemorrhage (HIPEC-No HIPEC)	Renal Complications (HIPEC-No HIPEC)	Cardiac Complications (HIPEC-No HIPEC
Primary Debulking Surgery							
Lei [28]	2020	2/425 (0.4%)–6/159(3.7%)	400/425 (94%)–142/159(89%)	3/425 (0.6%)–4/159 (2.5%)	2/425 (0.4%)–5/159(3.1%)	35/425 (8.2%)–7/159 (4.4%)	2/425 (0.4%)–3/159 (1.9%)
Lim [27]	2021	68/92 (73%)–82/92 (89%)	92/92 (100%)–92/92 (100%)	63/92 (68%)–69/92(75%)	3/92 (3%)–9/92(10%)	44/92 (47%)–63/92 (68%)	46/92 (50%)–48/92 (52%)
Ghirardi [22]	2022	NR	NR	NR	NR	NR	NR
Cascales Campos [20]	2014	NR	NR	NR	NR	NR	NR
Interval Debulking Surgery							
Mendivil [18]	2017	NR	22/69 (32%)	NR	10/69 (14%)	NR	NR
Ceresoli [29]	2018	NR	NR	NR	NR	NR	NR
Van Driel [12]	2018	52/122 (42%)–51/118 (43%)	7/122 (6%)–5/118 (4%)	13/122 (11%)–8/118 (7%)	4/122 (3%)–2/118 (2%)	NR	6/122 (5%)–8/118 (7%)
Cascales Campos [20]	2014	NR	NR	NR	NR	NR	NR
Batista [30]	2021	2/15 (13%)	5/15 (33%)	NR	1/15 (6%)	NR	NR
Marrelli [31]	2021	NR	10/46 (22%)	6/46 (13%)	2/46 (4%)	2/46 (4%)	1/46 (2%)
Lim [27]	2022	68/92 (73%)–82/92 (89%)	92/92 (100%)–92/92 (100%)	63/92 (68%)–69/92 (75%)	3/92 (3%)–9/92 (10%)	44/92 (47%)–63/92 (68%)	46/92 (50%)–48/92 (52%)
Wu [32]	2022	NR	NR	NR	NR	NR	NR
Recurrent Ovarian Cancer							
Spiliotis [14]	2014	NR	NR	NR	NR	NR	NR
Platinum sensitive disease	NR	NR	NR	NR	NR	NR	
Platinum resistant disease	NR	NR	NR	NR	NR	NR	
Cascales Campos [21]	2015	4/32 (12%)–2/22 (9%)	NR	1/32 (3%)–0	1/32 (3%)–1/22 (5%)	NR	NR
Baiocchi [28]	2016	NR	NR	NR	NR	NR	NR
Zivanovic [13]	2021	18/49 (37%)–32/49 (65%)	7/49 (14.3%)–8/49 (16.3%)	NR	1/49 (2%)–0	NR	NR

NR: not reported.

**Table 12 jcm-12-07012-t012:** HIPEC vs. no HIPEC complications *(part 2)*.

Author	Year	Infections (HIPEC-No HIPEC)	Performance Status (ECOG) 0 (HIPEC-No HIPEC)	Performance Status (ECOG) 1 (HIPEC-No HIPEC)	Performance Status (ECOG) 2 (HIPEC-No HIPEC)	Performance Status (ECOG) 3 (HIPEC-No HIPEC)
Primary Debulking Surgery						
Lei [28]	2020	56/425 (13%)–27/159 (17%)	NR	419/425 (98.6%)–156/159 (98.1%)	NR	6/425 (1.4%)–3/159 (1.9%)
Lim [27] (Total group)	2021	43/92 (47%)–47/92 (51%)	NR	NR	NR	NR
Ghirardi [22]	2022	NR	32/35 (91.4%)–32/35 (91.4%)	3/35 (8.6%)–3/35 (8.6%)	NR	NR
Cascales Campos [20] (Total group)	2014	NR	NR	48/52 (92%)–28/35 (80%)	4/52 (8%)–7/35 (20%)	NR
Interval Debulking Surgery						
Mendivil [18]	2017	NR	NR	NR	NR	NR
Ceresoli [29]	2018	NR	NR	NR	NR	NR
Van Driel [12]	2018	21/118 (18%)–14/122 (11%)	NR	NR	NR	NR
Cascales Campos [20] (Total group)	2014	NR	NR	48/52 (92%)–28/35 (80%)	4/52 (8%)–7/35 (20%)	NR
Batista [30]	2021	3/15 (20%)	3/15 (20%)	9/15 (60%)	3/15 (20%)	NR
Marrelli [31]	2021	NR	NR	NR	NR	NR
Lim [27] (Total group)	2022	43/92 (47%)–47/92 (51%)	NR	NR	NR	NR
Wu [32]	2022	NR	NR	3/46 (6.5%)–10/81 (12.4%)	43/46 (93.5%)–71/81 (87.7%)	NR
Recurrent Ovarian Cancer						
Spiliotis [14]	2014	NR	NR	NR	NR	NR
Platinum sensitive disease		NR	NR	NR	NR	NR
Platinum resistant disease		NR	NR	NR	NR	NR
Cascales Campos [21]	2015	3/32 (9%)–1/22 (5%)	NR	29/32 (91%)–19/22 (86%)	3/32 (9%)–3/22 (14%)	NR
Baiocchi [33]	2016	NR	NR	NR	NR	NR
Zivanovic [19]	2021	7/49 (14%)–3/49 (6%)	NR	NR	NR	NR

NR: not reported.

## Data Availability

The present review was based on published articles. All summary data generated during this study are included in this published article. Raw data used for the analyses are available and presented in the original reviewed articles.

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
