# Peer review of "Hyperthermic Intraperitoneal Chemotherapy (HIPEC): New Approaches and Controversies on the Treatment of Advanced Epithelial Ovarian Cancer—Systematic Review and Meta-Analysis"

_jcm, 2023, doi:10.3390/jcm12227012_

Round 1
Reviewer 1 Report
Comments and Suggestions for Authors
Citiations in Line 101 (17) and 188 (22) are wrong I suspect.
CHIPOR Trial should at least in the Discussion section be mentioned
In line 101: Please mention the trial name : which is randomizing newly diagnosed ovarian cancer patients .... the HORSE Trial would be for recurrent Ovarian cancer patients. Therefore Citation 22 would also be wrong...
The tables are complete but hard to understand for the reader, as the information in the tables is very important to compare the trials maybe there is an option to imporve these.
Comments on the Quality of English LanguageGood quality
Author Response
Citiations in Line 101 (17) and 188 (22) are wrong I suspect.
Response: Thanks for reporting the error, we have edited it.
CHIPOR Trial should at least in the Discussion section be mentioned
Response: As suggested, we have discussed it and the realtive reference (Lines 505-508 – Reference 42).
In line 101: Please mention the trial name: which is randomizing newly diagnosed ovarian cancer patients .... the HORSE Trial would be for recurrent Ovarian cancer patients. Therefore Citation 22 would also be wrong...
Response: Yes, as reported in the previous question, we have modified the citation involved since the whole discussion refers to [12] for the trial (M06OVH-OVHIPEC). Thank you for your support.
The tables are complete but hard to understand for the reader, as the information in the tables is very important to compare the trials maybe there is an option to improve these.
Response: your advice is surely appropriate. before to write this paper we have deeply reviewed all literature on this topic and according to other previous papers published on this topic and on the role of surgery in EOC, we have decided to set up the tables in the simplest and most complete way possible.
For this reason, the number of tables is high because the topic is complex and papers very heterogeneous. As suggested by the Reviewer, we modified the layout of the tables to be more clear and improve them.
Reviewer 2 Report
Comments and Suggestions for Authors
Dear all authors:
I would like to extend my congratulations to all of you for the completion of your intriguing work. The manuscript demonstrates significant potential. However, I have a few concerns and suggestions that I believe could enhance the quality of your research.
1. First of all, the clarity of content:
The manuscript could benefit from improved clarity in its content. The current format of paragraphing makes it challenging to understand the material effectively. I recommend revisiting the organization and structure of the text to ensure that the content is presented in a coherent and reader-friendly manner. Clear and concise paragraphs can significantly enhance the readability of the manuscript.
2. Discussion should focus on disparate outcomes:
The discussion section appears extensive but lacks a clear focus. Rather than delving into detailed descriptions of each study, it would be beneficial for the authors to concentrate on the underlying reasons for the disparate results or outcomes across the studies. By exploring these differences, you can contribute to a more insightful and focused discussion.
3. HIPEC controversies
Furthermore, given the significance of the topic, I recommend dedicating a portion of the discussion to the controversies surrounding HIPEC . This could involve an analysis of conflicting evidence, differences in clinical practice, or unresolved questions in the field. Addressing these controversies would add depth and relevance to your manuscript.
Comments on the Quality of English Language
Minor editing of English language required
Author Response
I would like to extend my congratulations to all of you for the completion of your intriguing work. The manuscript demonstrates significant potential. However, I have a few concerns and suggestions that I believe could enhance the quality of your research.
Response: Thnk you so much for your appreciation.
- First of all, the clarity of content:
The manuscript could benefit from improved clarity in its content. The current format of paragraphing makes it challenging to understand the material effectively. I recommend revisiting the organization and structure of the text to ensure that the content is presented in a coherent and reader-friendly manner. Clear and concise paragraphs can significantly enhance the readability of the manuscript.
Response: I understand your concern, but you have to take into account that we must respect the partition suggested by the Journal and the organization of subparagraph based on the topic: for example, the results section is divided in general characteristics, quality assessment, survival outcomes, PDS, IDS and recurrent OC because they report all aspect of this work such as all characteristics of patients included in the analysis, the aspects of studies included in the analysis and, last but not least, the 3 aspects of this review with metanalysis (PDS, IDS and recurrent). Regarding the discussion section, with have discussed the 3 aspects of this review, with the final analysis on strenghts and limitation. I hope in your comprehension. Thanks for the support.
- Discussion should focus on disparate outcomes:
The discussion section appears extensive but lacks a clear focus. Rather than delving into detailed descriptions of each study, it would be beneficial for the authors to concentrate on the underlying reasons for the disparate results or outcomes across the studies. By exploring these differences, you can contribute to a more insightful and focused discussion.
Response: As requested we have improved discussion section (Line 517-545). We hope we have satisfied your request.
- HIPEC controversies
Furthermore, given the significance of the topic, I recommend dedicating a portion of the discussion to the controversies surrounding HIPEC . This could involve an analysis of conflicting evidence, differences in clinical practice, or unresolved questions in the field. Addressing these controversies would add depth and relevance to your manuscript.
Response: Thank you for this important advise. As above reported, we have discussed this aspect (Lines 517-545).
According to your request, an English mother tongue revised the full text.